# OFFLINE REWARD INFERENCE ON GRAPH: A NEW THINKING

## ABSTRACT

In offline reinforcement learning, reward inference is the key to learning effective policies in practical scenarios. Due to the expensive or unethical nature of environmental interactions in domains such as healthcare and robotics, reward functions are rarely accessible, and the task of inferring rewards becomes challenging. To address this issue, our research focuses on developing a reward inference method that capitalizes on a constrained number of human reward annotations to infer rewards for unlabelled data. Initially, we leverage both the available data and limited reward annotations to construct a reward propagation graph, wherein the edge weights incorporate various influential factors pertaining to the rewards. Subsequently, we employ the constructed graph for transductive reward inference, thereby estimating rewards for unlabelled data. Furthermore, we establish the existence of a fixed point during several iterations of the transductive inference process and demonstrate its at least convergence to a local optimum. Empirical evaluations on locomotion and robotic manipulation tasks substantiate the efficacy of our approach, wherein the utilization of our inferred rewards yields substantial performance enhancements within the offline reinforcement learning framework, particularly when confronted with limited reward annotations.

## 1 INTRODUCTION

The offline reinforcement learning (RL) problem can be defined as a data-driven formulation of the reinforcement learning problem, that is, learning a policy from a fixed dataset without further environmental input (Lange et al., 2012; Levine et al., 2020; Prudencio et al., 2022). Reliable and effective offline RL methods would significantly affect various fields, including robots (Cabi et al., 2019; Dasari et al., 2019), autonomous driving (Yu et al., 2018), recommendation systems (Strehl et al., 2010; Bottou et al., 2013), and healthcare (Shortreed et al., 2011). Rewards are typically necessary for learning policies in offline RL, but they are rarely accessible in practice, and the rewards for state-action pairs need to be manually annotated, which is difficult and time-consuming. Meanwhile, real-world offline RL datasets always have a small amount with reward and a large amount always without reward. Thus, learning a model from limited data with rewards to label unrewarded data is critical for learning effective policies to apply offline RL to various applications.

Typical methods have attempted various types of supervision for reward learning. The method proposed by Cabi et al. (2020) and ORIL (Zolna et al., 2020) learn reward functions and use them in offline RL. Cabi et al. (2020) employs a reward sketching interface to elicit human preferences and use them as a signal for learning. In reward sketching, the annotator draws a curve where higher values correspond higher rewards. ORIL (Zolna et al., 2020) relies on demonstrated trajectories to obtain reward functions both from labelled and unlabelled data at the same time as training an agent. Konyushkova et al. (2020) propose the timestep annotations are binary and treat the reward prediction as a classification problem to focus on sample efficiency with limited human supervision.

Reward learning for offline RL is roughly divided into two categories: timestep-level (e.g., state-action pair reward annotations for the entire episode produced by humans (Cabi et al., 2020)) and episode-level supervision (e.g., annotations of success for the whole episode (Konyushkova et al., 2020; Zolna et al., 2020)). For episode-level supervision, Konyushkova et al. (2020) assumes rewards are binary, and it indicates if the task is solved. Episode annotations provide only limited information about the reward. They indicate that some of the state-action pairs from the episodes show successful

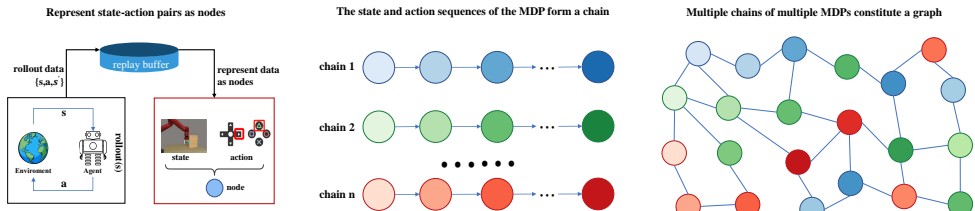

Figure 1: We first represent each state-action pair within a Markov decision process (MDP) as an individual graph node. Then, we establish a foundation for modeling the state-action sequences across multiple MDPs as interconnected chains. Finally, these chains collectively form a comprehensive graph that encapsulates the dynamics of multiple MDPs. The graph structure is characterized by connectivity, where each node is connected to multiple other nodes. We leverage this feature to propagate reward-related information within the graph.

behavior but do not indicate when the success occurs. So the episode-level supervision method is not adaptation to any value reward learning question. For timestep-level annotations, Konyushkova et al. (2020) and Yu et al. (2022) need first to annotate demonstrated trajectories that are the successful trajectories (e.g., expert demonstrations). Cabi et al. (2020) employs a reward sketching interface to elicit human preferences and use them as a signal for learning, but the method is hard to solve tasks where variable speed is important or with cycles as in walking. Accordingly, existing methods are unsuitable for timestep-level reward learning with arbitrary values in offline reinforcement learning without any expert trajectories. The manual annotation of rewards for state-action pairs is costly, making it challenging to learn effective policies with limited reward labelled data. This makes it challenging to apply offline reinforcement learning to a variety of scenarios.

In a general sense, offline RL addresses the problem of learning to control a dynamic system, which is fully defined by a Markov decision process (MDP). An MDP is a sequential decision process for state-to-state transitions, which could be formed as a chain, and multiple chains of multiple MDPs can be combined to form a graph. In the graph, each node represents a state-action pair, and each edge is labelled with the probability of transitioning from one state to another state, given a particular action. The graph structure is characterized by connectivity, where each node is connected to multiple other nodes. It is convenient to perform inference through a message passing mechanism, and the inference procedure essentially propagates information from the state-action pairs with rewards to the state-action pairs without rewards and results in a reward function over state-action pairs, as shown in Figure 1. Meanwhile, the label-efficiency problem has been successfully addressed through label propagation (LPA) approaches (Zhu & Ghahramani, 2002; Zhu, 2005; Zhou et al., 2003; Zhang & Lee, 2006; Wang & Zhang, 2006; Karasuyama & Mamitsuka, 2013; Gong et al., 2016; Liu et al., 2018) which is based on graph models and plays a significant role in leveraging unlabelled dataset to improve the model performance with low cost. Label propagation approaches formulate labelled and unlabelled data as a graph structure, where nodes represent sample data and edges represent relationships between nodes, and the node labels are propagated and aggregated along the edges.

Inspired by the label propagation approaches, we present TRAIN: **T**ransductive **R**ew**A**rds **IN**ference with Propagation Graph for Offline Reinforcement Learning. Transduction is reasoning from observed training cases to test cases. We learn from the already observed state-action pairs with rewards and then predict the state-action pairs without rewards. Even though we do not know the rewards of most state-action pairs, we can leverage the graph structure established by MDPs to propagate limited reward information. To be specific, TRAIN consists of the following key ingredients:

(a) Reward Propagation Graph: We represent each state-action pair as a graph node and leverage the similarities and relationships between them to learn the edge weights of the nodes. It is worth noting that rewards are influenced by many factors, and all of the factors should be considered when learning the reward propagation graph.

(b) Transductive Reward Inference: We employ the reward propagation graph to infer rewards for state-action pairs that are without rewards and then utilize them for doing offline RL. The reward inference technique propagates rewards on the graph from state-action pairs with rewards to state-action pairs without rewards and will converge to a unique fixed point after a few iterations.

We remark that TRAIN is not a naïve application of the LPA technique but a novel scalable method of learning propagation graphs that integrates multiple influence reward factors to edge weights. The graph sufficiently leverages various relationship information between nodes, which can make reward inference more accurate. This has not been considered or evaluated in the context of offline RL reward learning. We also prove that the transductive inferred reward has a fixed point and at least can converge to a local optimum.

Our experiments demonstrate that the state-action pairs labeled by TRAIN significantly improve the offline reinforcement learning method when learning policy with limited reward annotations on complex locomotion and robotic manipulation tasks from DeepMind Control Suite (Tassa et al., 2018) and Meta-World (Yu et al., 2020). In particular, our method inherits the smooth characteristics of the LPA method (Wang & Leskovec, 2020), which can make the state-action pairs with smooth rewards and further make the process of offline RL algorithm learning policy more stable.

## 2 RELATED WORK

**Offline RL** The offline reinforcement learning problem, which enables learning policies from the logged data instead of collecting it online, can be defined as a data-driven formulation of the reinforcement learning problem (Lange et al., 2012; Levine et al., 2020; Prudencio et al., 2022). It is a promising approach for many real-world applications. Offline RL is an active area of research and many algorithms have been proposed recently, e.g., BCQ (Fujimoto et al., 2019), MARWIL (Wang et al., 2018), BAIL (Chen et al., 2020), ABM (Siegel et al., 2020), AWR (Peng et al., 2019), CRR (Wang et al., 2020), F-BRC (Kostrikov et al., 2021). In this paper, we adopt CRR as our backbone algorithm due to its efficiency and simplicity.

**Reward learning** It is possible to learn the reward signal even when it is not constantly available in the environment. The reward can be learned if demonstrations are provided either directly with inverse RL (Abbeel & Ng, 2004; Ng et al., 2000) or indirectly with generative adversarial imitation learning (GAIL) (Ho & Ermon, 2016). The end goal (Edwards et al., 2016; Singh et al., 2019) or reward values (Cabi et al., 2020) for a subset of state-action pairs can be known, in which case reward functions can be learned by supervised learning. A significant instance of learning via limited reward supervision (Cabi et al., 2020) is studied in some works. Rewards are commonly learned for online RL (Klissarov & Precup, 2020). While learning from built or pre-trained state representations (Baram et al., 2017; Edwards et al., 2016; Finn et al., 2016; Fu et al., 2017; Li et al., 2017; Merel et al., 2017; Sermanet et al., 2016; Zhu et al., 2018) has achieved a lot of success, learning directly from pixel input is known to be difficult (Zolna et al., 2021) and the quantity of supervision needed may become a bottleneck (Cabi et al., 2020). Unlike many other reward learning approaches for offline RL, we focus on learning rewards with multi-factors that influenced rewards from limited annotations.

**Transduction** The setting of transductive inference was first introduced by Vapnik (Vapnik, 1999). Transductive Support Vector Machines (TSVMs) (Joachims et al., 1999) is a margin-based categorization technique that reduces test set mistakes. Particularly for short training sets, it demonstrates considerable advantages over inductive techniques. Another classification of transduction methods involves graph-based methods (Zhu & Ghahramani, 2002; Zhou et al., 2003; Wang & Zhang, 2006; Rohrbach et al., 2013; Fu et al., 2015). Labels are transferred from labelled to unlabelled data instances through a process called label propagation, which is driven by the weighted graph. In prior works, the graph construction is done on a pre-defined feature space using only a single influence factor between nodes so that it is not possible to learn multi-factors influenced graph edge weights.

## 3 PROBLEM FORMULATION

The key to the TRAIN method is the prior assumption of consistency, which means: (1) nearby states and actions are likely to have similar or the same reward, and (2) state-action pairs on the same structure (typically referred to as a cluster or a manifold) are likely to have the similar or the same reward. This argument is akin to semi-supervised learning problems that in (Belkin & Niyogi, 2002; Blum & Chawla, 2001; Chapelle et al., 2002; Zhou et al., 2003; Zhu et al., 2003; Wang & Leskovec, 2020; Iscen et al., 2019) and often called the *cluster assumption* (Zhou et al., 2003; Chapelle et al., 2002). Orthodox supervised learning algorithms, such as k-NN, in general, depend only on the first

assumption of local consistency (Zhou et al., 2003), that is, k-NN makes every data point be similar to data points in its local neighborhood. Our method leverages the relation information between states and actions to formalize the intrinsic structure revealed by state-action pairs with reward and state-action pairs without reward and construct a reward inference function.

We assume that the training samples (both with reward and without reward) are given as $D := [D_L, D_U] = [(s_1, a_1), ..., (s_L, a_L), (s_{L+1}, a_{L+1}), ..., (s_U, a_U)]$, where $(s_i, a_i)$ denotes the state-action pair. The first $L$ samples $(s_i, a_i)$ for $i \in L := 1, ..., L$, denoted by $D_L$, are with reward. The remaining $U$ samples for $i \in U := L + 1, ..., U$, denoted by $D_U$, are without reward. $A = |L| + |U|$ is the total number of training samples. The rewards are denoted by $R := [R_L, R_U] = [r_1, ..., r_L, r_{L+1}, ..., r_U]$. Suppose that we are given a small set $D_L$ of the state-action pairs with reward. The rest of the state-action pairs $D_U = D \setminus D_L$ is without reward. TRAIN utilize all samples and known rewards to learn a reward propagation graph and infer rewards for state-action pairs that are without reward.

## 4 METHODOLOGY

### 4.1 OVERVIEW

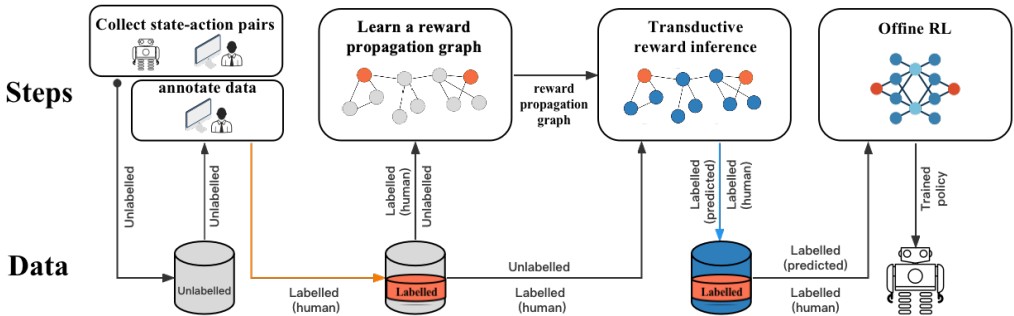

Figure 2: TRAIN workflow: We first learn a reward propagation graph on a pre-recorded dataset. Then, we employ the graph and state-action pairs with rewards to infer rewards for state-action pairs without rewards. Finally, we leverage all state-action pairs to do offline RL.

We take advantage of the property of the Markov Decision Process (MDP) that the reward function on an MDP depends only on the current state and action. Therefore, we leverage the relationship between states and actions to learn the reward propagation graph and realize reward labeling for state-action pairs without rewards. As in offline Reinforcement Learning, such state-action pairs are logged in a dataset D. In practice, D includes diverse state-action pairs produced for various tasks by scripted, random, or learned policies as well as human demonstrations (Cabi et al., 2020).

### 4.2 CONSTRUCT REWARD PROPAGATION GRAPH

The graph structure is characterized by connectivity and can model the interrelationships between entities. In a graph, edges connect nodes, and the information of nodes can be transmitted to other nodes through the connection relationship of edges. We need to use the limited state-action pairs with rewards to learn rewards for the state-action pairs without rewards. Therefore, we model state-action pairs as nodes and learn a reward propagation graph using the relationships between nodes, which transfers information from labelled reward nodes to unlabelled reward nodes.

**Graph construction**  For most reinforcement learning tasks, rewards are influenced by many factors. For instance, in task *Humanoid*, which is part of the DeepMind Control Suite (Tassa et al., 2018; Tunyasuvunakool et al., 2020), the state consists of six parts: joint angles, the height of the torso, extremity positions, torso vertical orientation, the velocity of the center of mass, and the generalized velocity, and action also consists of several parts that represent the torques applied at the hinge joints. The reward is related to the upright state of the robot, the control operation of the actuator, and the moving speed, etc., which are closely related to each part of the above state and action, so we regard each part of the state and action as a factor that influences the reward.

Specifically, we denote $s_i = [s_{i_1}, s_{i_2}, s_{i_3}, ..., s_{i_M}]$, where $s_{i_m}$ is a sub-state with any given dimension, and $s_i$ consists of $M$ sub-states. Correspondingly, we denote $a_i = [a_{i_1}, a_{i_2}, a_{i_3}, ..., a_{i_N}]$, where $a_i$ is all of the actions performed given state $s_i$, and composed of $N$ specific sub-actions of $a_{i_n}$.

We design a reward propagation graph construction method integrating multi-factors influencing the reward to tune the edge weights. To be specific, we employ a distance function $\mathcal{D}_s(s_{i_m}, s_{j_m}), \forall\, i \neq j$ to measure the similarity of the sub-states, and also employ a distance function $\mathcal{D}_a(a_{i_n}, a_{j_n}), \forall\, i \neq j$ to measure the similarity of the sub-actions, the two distance function could be Euclidean distance or others. Then, we define the multi-factor measure for state-action pairs as:

$$\mathcal{D}((s_i, a_i), (s_j, a_j)) = [\mathcal{D}_s(s_{i_1}, s_{j_1}), ..., \mathcal{D}_s(s_{i_M}, s_{j_M}), \mathcal{D}_a(a_{i_1}, a_{j_1}), ..., \mathcal{D}_a(a_{i_N}, a_{j_N})] \in \mathbb{R}^{M+N}. \tag{1}$$

Further, we employ a reward shaping function $f_\Theta$ with the parameters $\Theta$ to tune multi-factors' contribution to the rewards and integrate them into the edge weight:

$$W_{ij} = \frac{exp(-f_\Theta(\mathcal{D}((s_i, a_i), (s_j, a_j))))}{\sum_{j \neq i} exp(-f_\Theta(\mathcal{D}((s_i, a_i), (s_j, a_j))))}, \forall\, i \neq j. \tag{2}$$

where $W_{ij}$ is an element in matrix $W$, which is a $(L + U) \times (L + U)$ weight matrix. We let $W_{ii} = 0$ and we also have $\sum_j W_{ij} = 1, \forall i$. We define each state-action pair as a node in the graph and define the weight between every two nodes in the graph, thus completing the construction of the reward propagation graph $G$ with weight matrix $W$.

**Graph training**   In different tasks, the number of factors influencing reward is different, and the degree of influence of different factors on reward is also different. Therefore, we need to tune the weight of graph edges so that optimizing the function $f_\Theta$ to make the multi-factors efficiently integrate to rewards. Intuitively, for the state-action pair $(s_i, a_i)$, a larger edge weight $W_{ij}$ means that state-action pair $(s_j, a_j)$ will transfer more information to the reward for state-action pair $(s_i, a_i)$. Specifically, we use the relationship between labelled data to train $f_\Theta$ to make it suitable for the current task. We design a predicted reward $\xi_l$:

$$\xi_l = \sum_{k \neq l} W_{lk} r_k, \quad k, l \in [1, ..., L], \tag{3}$$

where $r_k$ is a label (reward) for a state-action pair (node). We use other labelled state-action pairs to predict the label of the current state-action pair (have a ground truth label) and then minimize the difference between the predicted label and the ground truth label.

Then, the goal of training the graph $G$ is to optimize the parameters $\Theta$ for the function $f_\Theta$, that is, minimize the difference between the predicted labels and their corresponding ground truth labels, the objective function $H(G)$ is given as:

$$\arg\min_{\Theta} \left\{ H(G) = \frac{1}{2L} \sum_{l=1}^{L} ||\xi_l - r_l||^2 \right\}. \tag{4}$$

There does not exist a closed-form solution, and we use the gradient descent method to seek the solution, details are in the appendix A.1.

It should be noted that: the unlabelled data are not included in Equation (4), since the number of the unlabelled data is often much larger than that of the labelled data, the term on the unlabelled data may dominate the objective function, which in turn may degrade the algorithmic performance.

### 4.3   TRANSDUCTIVE REWARD INFERENCE

In Section 4.2, we constructed the graph and trained the weights of the graph edges. In this section, we propagate reward-related information on the graph to infer the rewards for unlabelled state-action pairs based on the rewards of other state-action pairs.

We separate the weights associated with nodes without rewards from the weight matrix $W$ of graph $G$ formed by Equation (2), and representing them as two submatrices $W_{UL}$ and $W_{UU}$. $W_{UL}$ represents

the weights between nodes with rewards and nodes without rewards, and $W_{UU}$ represents the weights between nodes without rewards. We also split the reward set $R$ into 2-sub-block $R = [R_L, R_U]$, where $R_L$ denotes the subset of known rewards and $R_U$ denotes the subset of unknown rewards.

The inference of rewards for unlabelled nodes requires considering the information transfer between labelled and unlabelled nodes, we use $W_{UL}R_L$ for this calculation, while also taking into account the relationships between unlabeled nodes themselves, computed using $W_{UU}R_U$. Since unknown rewards $R_U$ are a variable to be learned, we provide an iterative computation formula by:

$$R_U \longleftarrow W_{UU}R_U + W_{UL}R_L. \tag{5}$$

After t-th iterations, we obtain the following formula:

$$R_U^t \longleftarrow W_{UU}^t R_U^0 + (W_{UU}^{t-1} + ... + W_{UU} + 1)W_{UL}R_L. \tag{6}$$

The detailed derivation process is in Appendix A.2. Based on the weights calculated by Equation (2), the values in $W_{UU}$ are all less than 1. Therefore, as $t$ approaches infinity, $W_{UU}^t$ tends to infinitesimal values, leading $W_{UU}^t R^0$ converges to 0. Meanwhile, $(W_{UU}^{t-1} + ... + W_{UU} + 1)$ forms a geometric series, and after applying the formula for the sum of a geometric series, we obtain the solution:

$$R_U = (I - W_{UU})^{(-1)}W_{UL}R_L, \tag{7}$$

which is a *fixed point*, and $I$ is the identity matrix (Zhu & Ghahramani, 2002; Zhou et al., 2003). TRAIN converges to a *fixed point* means that the reward inference error is within a certain range.

### 4.4 POLICY LEARNING

We remark that TRAIN can be combined with any offline RL algorithm by learning rewards for state-action pairs without reward. For learning a policy, we use a pre-recorded dataset D. Dataset D contains some state-action pairs with reward $D_L$, and most of the rest are state-action pairs without reward $D_U$. We use TRAIN to predict rewards for $D_U$ as $\tilde{D}_U$, and form $\tilde{D}$ with D and $\tilde{D}_U$, then do offline RL with $\tilde{D}_L$. In the paper, we employ Critic-Regularized Regression (CRR) (Wang et al., 2020), a simple and efficient offline reinforcement learning algorithm, to train an offline reinforcement learning policy on the dataset with predicted rewards.

## 5 EXPERIMENTS

### 5.1 EXPERIMENTS SETUP

**Environment and tasks** We conduct the experiments with a variety of complex robotic manipulation and locomotion tasks from Meta-World (Yu et al., 2020) and DeepMind Control Suite (Tassa et al., 2018; Tunyasuvunakool et al., 2020), respectively. Many factors influence the reward function of these two series of tasks. Meta-World consists of a variety of manipulation tasks designed for learning diverse manipulation skills. The second environment is the DeepMind Control Suite, which contains many continuous control tasks involving locomotion and simple manipulation. To investigate the performance of TRAIN with a small amount of annotated state-action pairs. We conduct experiments on more than 40 tasks in the Meta-World environment, and we select four tasks (*Hammer, Button Press, Sweep Into, Open Drawer*) (see Figure 3) of them to show

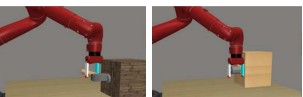

(a) Hammer   (b) Button Press

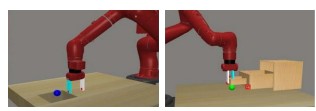

(c) Sweep Into (d) Open Drawer

Figure 3: Meta-World is a set of robotic manipulation tasks.

their learning curves. We also choose five complex environments from DeepMind Control Suite: *Cheetah Run, Walker Walk, Fish Swim, Humanoid Run, and Cartpole Swingup* (see Figure 4) to evaluate the performance of TRAIN in another environment different from the Meta-World, to verify whether the algorithm works only in one environment. On each task, we leverage different algorithms to predict the rewards for the same dataset and employ the CRR algorithm (except Behavior cloning) to perform policy learning on the dataset after predicting the rewards. The learned policies are used

to evaluate the performance of TRAIN and other baselines. We report the results on all tasks of 5 random seeds, and results are shown in Section 5.2.

**Datasets** Given a set of logged state-action pairs (dataset D), we random extract a small subset of state-action pairs and leverage them as labelled data $D_L$, and others as $D_U$. The labels of $D_L$ are the ground truth rewards. Meta-World domain has been used to evaluate online RL agents, we create an *ad hoc* dataset suitable for offline learning. To do so, we train

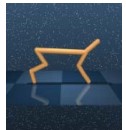 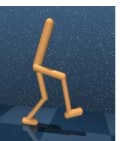 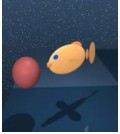 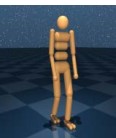 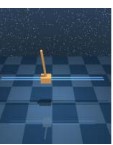

(a) Cheetah   (b) Walker   (c) Fish   (d) Humanoid (e) Cartpole

Figure 4: DeepMind Control Suite is a set of popular continuous control environments with tasks of varying difficulty, including locomotion and simple object manipulation.

Soft Actor-Critic (Haarnoja et al., 2018) from full states on each of the tasks, and save the resulting replay buffer, which forms the dataset D. For DeepMind Control Suite, we use the open source RL Unplugged datasets (Gulcehre et al., 2020) to form the dataset D for each task, the dataset also contains both successful and unsuccessful episodes. The total numbers of state-action pairs and the proportion of state-action pairs with ground truth rewards for each task are shown in Appendix A.3.

**Baselines** Behavior cloning (**BC**) is a popular algorithm in the field of imitation learning and also an alternative way to learn a policy when the reward values are not available. BC agent does not require reward values as it attempts to directly imitate the demonstrated state-action pairs. Time-guided reward (**TGR**) (Konyushkova et al., 2020) adopts a two-step approach by initially annotating demonstrated trajectories and a flat zero synthetic reward is assigned to the unlabelled subset. The reward function is then trained using a loss function that is optimized jointly over timestep-level annotations and synthetic flat labels. We also combined TGR with CRR (Wang et al., 2020) to learn policies. Unlabelled data sharing (**UDS**) (Yu et al., 2022) assigns the lowest possible reward (usually assumed to be 0) to all transitions in the unlabelled data. Subsequently, the unlabelled transitions are reweighted to change the distribution of unlabelled data, aiming to mitigate reward bias.

## 5.2 EXPERIMENTS RESULTS

**Meta-world** We show the evaluation results on more than 40 tasks in the Meta-World environment in Table 1. Apart from the five tasks (*button-press, drawer-close, handle-press, reach, reach-wall*), the performance of the algorithm TRAIN exhibits greater superiority compared to others. This indicates that TRAIN achieves more accurate reward learning on these tasks. Additionally, due to its inherent smoothness, the learned rewards in TRAIN are smoother, leading to more stable policy performance. UDS emphasizes the need for high-quality labeled data. Since the data in our environment consists of SAC training data, and the labeled data is randomly selected, UDS performs well in relatively easy-to-learn tasks, specifically those with a higher proportion of high-quality data in the dataset, with particularly outstanding performance in the tasks of *button-press, drawer-close, and handle-press*. But, its performance is subpar in many other tasks.

Regarding the two tasks where the TGR algorithm outperforms others, we conducted a detailed analysis to identify the reasons. These tasks involve relatively simple action trajectories that are easily explored. The TGR algorithm annotates the demonstrated trajectories and assigns a flat zero synthetic reward to the unlabelled subset, which amplifies the rewards annotated as one along the action trajectories, encouraging the policy to learn these trajectories more actively. Therefore, TGR achieves better performance on such tasks. However, in other relatively complex tasks where the action trajectories for task completion are diverse, the method fails to provide accurate rewards for some procedural actions, resulting in mediocre policy performance. On the other hand, the poor performance of the Behavioral Cloning (BC) algorithm in many tasks can be attributed to its reliance on training with the entire dataset. The dataset comprises both successful and unsuccessful episodes, causing BC to be significantly affected by data quality. In tasks with relatively simple actions, algorithms that collect data can quickly learn the action trajectories for task completion, leading to a higher proportion of high-quality data in the dataset and thus better performance of BC. Conversely, in relatively complex tasks, data collection algorithms require a longer exploration process to learn the action trajectories for task completion, resulting in a lower proportion of high-quality data and consequently poor performance of BC.

Table 1: Evaluation returns on more than 40 Meta-World tasks. The average ± standard deviation is shown for five random seeds. Italic numbers indicate the highest average return for each task. Bold numbers indicate the statistically significant highest average return for each task, in which the average return is the highest, and there is not a clear overlap in the standard deviation among the baselines.

| Tasks | BC | TGR | UDS | TRAIN |
|---|---|---|---|---|
| basketball | $3019 \pm 439.6$ | $978 \pm 122.7$ | $3661 \pm 406.9$ | $4165 \pm 315.9$ |
| box-close | $1081 \pm 106.4$ | $1568 \pm 119.8$ | $1371 \pm 134.5$ | $\mathbf{3897 \pm 412.3}$ |
| button-press | $1911 \pm 356$ | $3301 \pm 291$ | $3508 \pm 213.4$ | $3435 \pm 106.4$ |
| button-press-topdown | $3190 \pm 310.7$ | $3401 \pm 415.7$ | $3496 \pm 229.2$ | $3587 \pm 75.4$ |
| button-press-topdown-wall | $1892 \pm 50.2$ | $2085 \pm 61.6$ | $2069 \pm 48.7$ | $2115 \pm 70.4$ |
| coffee-button | $3531 \pm 710.6$ | $3631 \pm 610.8$ | $3923 \pm 352.4$ | $4128 \pm 210.9$ |
| coffee-pull | $517 \pm 16.3$ | $372 \pm 11.5$ | $314.2 \pm 7.9$ | $\mathbf{637 \pm 35.2}$ |
| dial-turn | $1871 \pm 262.7$ | $4217 \pm 395.4$ | $4236 \pm 162.1$ | $4428 \pm 185.9$ |
| disassemble | $215 \pm 10.9$ | $217 \pm 18.1$ | $239.7 \pm 21.5$ | $\mathbf{889 \pm 10.5}$ |
| door-close | $3862 \pm 167.1$ | $4328 \pm 212.0$ | $4451 \pm 184.2$ | $4512 \pm 217.3$ |
| door-lock | $3156 \pm 306.9$ | $3536 \pm 285.1$ | $3312 \pm 182.8$ | $3753 \pm 195.2$ |
| door-open | $1082 \pm 46.7$ | $3985 \pm 306.9$ | $1949 \pm 66.1$ | $4451 \pm 197.1$ |
| door-unlock | $1947 \pm 216.1$ | $4011 \pm 75.3$ | $2052 \pm 101.6$ | $4189 \pm 118.9$ |
| drawer-close | $4697 \pm 64.3$ | $4839 \pm 102.1$ | $4881 \pm 79.8$ | $4797 \pm 20.3$ |
| drawer-open | $1769 \pm 247$ | $2890 \pm 86$ | $1895 \pm 40.1$ | $\mathbf{4466 \pm 41.3}$ |
| faucet-close | $4143 \pm 170.6$ | $4687 \pm 821.6$ | $4338 \pm 102.1$ | $4712 \pm 147.1$ |
| faucet-open | $3660 \pm 316.4$ | $4702 \pm 1503.5$ | $4671 \pm 361.8$ | $4715 \pm 361.3$ |
| hammer | $2205 \pm 268$ | $3898 \pm 163$ | $2692 \pm 101.5$ | $\mathbf{4532 \pm 95.1}$ |
| hand-insert | $56 \pm 16.1$ | $443 \pm 8.4$ | $409 \pm 10.8$ | $\mathbf{4016 \pm 598.6}$ |
| handle-press | $4598 \pm 137.9$ | $4522 \pm 136.4$ | $4651 \pm 82.4$ | $4618 \pm 102.8$ |
| handle-press-side | $4241 \pm 1353.4$ | $4764 \pm 243.7$ | $4734 \pm 185.2$ | $4785 \pm 458.1$ |
| handle-pull | $3892 \pm 986.8$ | $4348 \pm 894.1$ | $4138 \pm 147.6$ | $4592 \pm 125.7$ |
| handle-pull-side | $3678 \pm 1006.2$ | $4095 \pm 572.6$ | $3958 \pm 114.9$ | $\mathbf{4551 \pm 92.8}$ |
| lever-pull | $3864 \pm 190.8$ | $4184 \pm 175.1$ | $4008 \pm 81.7$ | $\mathbf{4307 \pm 135.7}$ |
| pick-out-of-hole | $11 \pm 0.4$ | $209 \pm 3.5$ | $117 \pm 12.2$ | $\mathbf{1035 \pm 234.9}$ |
| pick-place | $1879 \pm 411.6$ | $2975 \pm 495.6$ | $3228 \pm 283.2$ | $\mathbf{4106 \pm 589.4}$ |
| plate-slide | $3984 \pm 101.7$ | $3674 \pm 748.3$ | $4064 \pm 151.8$ | $4459 \pm 171.4$ |
| plate-slide-back | $3017 \pm 331.6$ | $3158 \pm 958.4$ | $3089 \pm 163.1$ | $\mathbf{4658 \pm 165.3}$ |
| plate-slide-back-side | $4087 \pm 887.5$ | $4678 \pm 172.6$ | $4703 \pm 136.8$ | $4734 \pm 198.4$ |
| plate-slide-side | $2698 \pm 538.8$ | $3002 \pm 365.3$ | $2928 \pm 289.1$ | $3010 \pm 429.5$ |
| push | $1983 \pm 381.9$ | $2097 \pm 261.4$ | $2248 \pm 175.2$ | $\mathbf{4268 \pm 210.7}$ |
| push-back | $9 \pm 0.4$ | $137 \pm 1.5$ | $79 \pm 8.7$ | $\mathbf{201.3 \pm 21.7}$ |
| push-wall | $3642 \pm 597.8$ | $4347 \pm 187.4$ | $4182 \pm 155.8$ | $4501 \pm 204.6$ |
| reach | $3209 \pm 397.2$ | $4761 \pm 476.6$ | $4581 \pm 181.2$ | $4668 \pm 215.6$ |
| reach-wall | $4626 \pm 91.8$ | $4816 \pm 51.1$ | $4672 \pm 30.4$ | $4810 \pm 36.3$ |
| stick-pull | $592 \pm 10.8$ | $442 \pm 7.2$ | $408 \pm 7.8$ | $\mathbf{4128 \pm 121.5}$ |
| stick-push | $362 \pm 16.2$ | $887 \pm 5.3$ | $1065 \pm 21.7$ | $\mathbf{2745 \pm 514.3}$ |
| sweep | $879 \pm 145.6$ | $3214 \pm 412.7$ | $3708 \pm 237.1$ | $4106 \pm 312.7$ |
| sweep-into | $962 \pm 137$ | $1838 \pm 149$ | $2115 \pm 176.5$ | $2257 \pm 323.9$ |
| window-close | $3846 \pm 98.3$ | $4104 \pm 106.1$ | $4354 \pm 86.4$ | $4458 \pm 88.4$ |
| window-open | $3217 \pm 193.2$ | $2897 \pm 954.5$ | $3141 \pm 105.5$ | $\mathbf{3829 \pm 208.4}$ |

We select four tasks of the Meta-World (*Hammer, Button Press, Sweep Into, Open Drawer*) to show their learning curves as measured on the success rate, as shown in Figure 5. TRAIN outperforms the baselines on all four tasks, showing that TRAIN is well suited to make effective use of the unlabelled, mixed quality, state-action pairs. In the two tasks of Hammer and Sweep Into, TRAIN has always shown a greater performance advantage compared to other baselines. In the two tasks of Button Press and Open Drawer, in the early stage of training, the performance of the three algorithms is equivalent, and the follow-up TRAIN gradually stands out. Especially in the Open Drawer task, a greater performance advantage has been achieved, while in the Button Press task, the training process of other baselines fluctuates greatly, and TRAIN has less fluctuation and achieves better performance. The conclusions drawn from the results in Figure 5 are consistent with the conclusions drawn from the results in Table 1, and TRAIN has an excellent performance in the success rate showing that

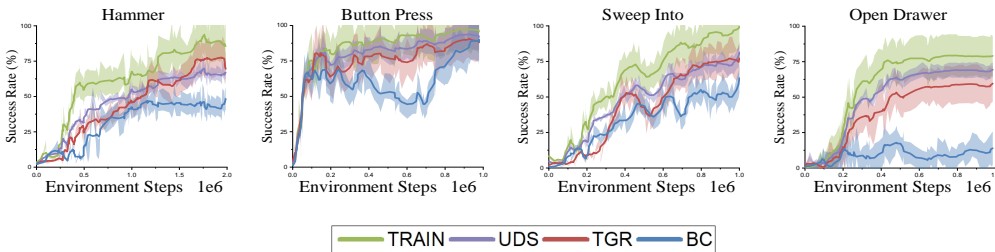

Figure 5: Learning curves on the four Meta-World tasks as measured on the success rate. The solid line and shaded regions represent the mean and standard deviation, respectively, across five seeds.

our algorithm TRAIN can effectively label rewards for state-action pairs without rewards, and the smoothness of the algorithm can make the labelled data also have smooth characteristics. The policies trained using these data have more stable performance.

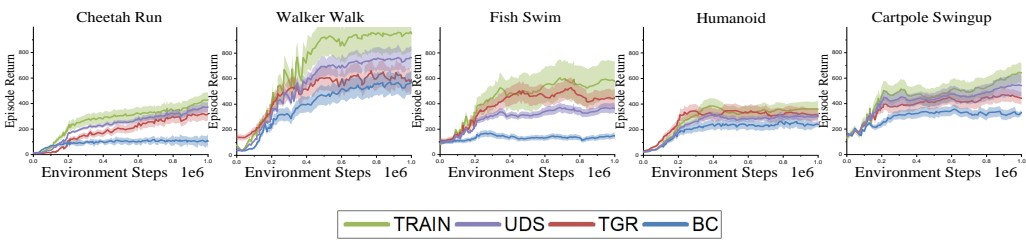

Figure 6: Learning curves on the five DM Control tasks as measured on the episode return. The solid line and shaded regions represent the mean and standard deviation, respectively, across five seeds.

**DeepMind Control Suite**     We show the learning curves of the five DeepMind Control Suite tasks (*Cheetah Run, Walker Walk, Fish Swim, Humanoid, Cartpole Swingup*) in Figure 6. Our method TRAIN achieves better performance than baselines in five tasks. It has achieved a great performance advantage in the Walker Walk task, and also performed well in the Fish Swim task. In the Cheetah Run and Cartpole Swingup tasks, it also showed a certain performance advantage compared to TGR, although the advantage is not very large, but it can still reflect the ability of the TRAIN algorithm. In the Humanoid task, TRAIN, UDS and TGR are evenly matched during the training process, but the final performance of TRAIN is still better than other baselines, reflecting the stability of the state-action pairs with rewards provided by the TRAIN algorithm.

The action spaces of these five tasks are continuous, and the reward is also continuous. It is difficult to give a clear boundary to distinguish good actions and bad actions. Therefore, the performance of TRAIN and baselines are both very steady. However, the TRAIN algorithm has smooth characteristics, which can make the labelled data also have smooth characteristics. Using these state-action pairs with smooth rewards makes the process of offline RL algorithm learning policy more stable. Since the TGR algorithm annotates the demonstrated trajectories and assigns a flat zero synthetic reward to the unlabelled subset, it shows a large shock in the process of learning some tasks. UDS emphasizes the need for high-quality labeled data, resulting in slightly poorer performance. BC performed the worst among the five tasks, and hardly worked in the two tasks of Fish Swim and Cheetah Run. Because BC uses a complete data set for training, it cannot distinguish the quality of the data, and cannot obtain a policy with excellent performance.

# 6    CONCLUSION

In conclusion, our research propose TRAIN method to addresses a critical challenge in offline reinforcement learning by developing a reward inference method that leverages a constrained number of human reward annotations to estimate rewards for unlabelled data. TRAIN model MDPs as a graph and leverage the connectivity of the graph structure to constructs a reward propagation graph that incorporates various influential factors, facilitating transductive reward inference. We have shown the existence of a fixed point during the iterative inference process, and our method converges at least to a local optimum. Empirical evaluations on locomotion and robotic manipulation tasks demonstrate the effectiveness of TRAIN, especially when dealing with limited reward annotations. This work has significant implications for practical scenarios where reward functions are challenging to access.

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

# A  APPENDIX

## A.1  GRADIENT

For the graph $G$, we design the training loss function $H(G)$ in Equation (4), there does not exist a closed-form solution, and we use the gradient descent method to seek the solution. The gradient with respect to $\Theta$ is given in Equation (8),

$$
\begin{aligned}
\frac{\partial H}{\partial \Theta} &= -\frac{1}{L}\sum_l (r_l - \xi_l)\frac{\partial \xi_l}{\partial \Theta} \\
&= -\frac{1}{L}\sum_l (r_l - \xi_l)(-2\Theta)\left\{ [\sum_{k\neq l} r_k\xi_{lk}(\mathcal{D}_s(s_l,s_k)+\mathcal{D}_a(a_l,a_k))^2] \right. \\
&\quad \left. - [\sum_{k\neq l} r_k\xi_{lk}][\sum_{k\neq l}\xi_{lk}(\mathcal{D}_s(s_l,s_k)+\mathcal{D}_a(a_l,a_k))^2]\right\} \\
&= \frac{2\Theta}{L}\sum_l (r_l-\xi_l)\left\{ [\sum_{k\neq l} r_k\xi_{lk}(\mathcal{D}_s(s_l,s_k)+\mathcal{D}_a(a_l,a_k))^2] \right. \\
&\quad \left. -[\xi_l][\sum_{k\neq l}\xi_{lk}(\mathcal{D}_s(s_l,s_k)+\mathcal{D}_a(a_l,a_k))^2]\right\} \\
&= \frac{2\Theta}{L}\sum_l (r_l-\xi_l)\left\{ [\sum_{k\neq l} r_k\xi_{lk}(\mathcal{D}_s(s_l,s_k)+\mathcal{D}_a(a_l,a_k))^2] \right. \\
&\quad \left. -[\sum_{k\neq l}\xi_l\xi_{lk}(\mathcal{D}_s(s_l,s_k)+\mathcal{D}_a(a_l,a_k))^2]\right\} \\
&= \frac{2C}{L}\sum_{k,l} (r_l-\xi_l)(r_k-\xi_l)\xi_{lk}(\mathcal{D}_s(s_l,s_k)+\mathcal{D}_a(a_l,a_k))^2,
\end{aligned}
\tag{8}
$$

where $\Theta$ is the parameters of the function $f_\Theta$, $r_l$ is a label (reward) for a state-action pair (node), $\xi_l$ is a predicted label, and we attribute the constant term to $C$, the goal of training the graph $G$ is to minimize the difference between the predicted labels and their corresponding ground truth labels.

The primary objective is to minimize the difference between the predicted labels $\xi_l$ and their corresponding ground truth labels $r_l$ for state-action pairs (nodes) within the graph $G$. The equation calculates how changes in the model parameters $\Theta$ affect the loss. It starts with a summation symbol, which implies that the gradient is computed by summing up certain terms for all possible $l$ values. Difference between predicted and true labels: The term $(r_l - \xi_l)$ represents the difference between the true label $r_l$ and the predicted label $\xi_l$ for a specific state-action pair $l$. The next part, $\frac{\partial \xi_l}{\partial \Theta}$, represents the derivative of the predicted label $\xi_l$ with respect to the model parameters $\Theta$. This part shows how small changes in the parameters affect the predicted label. These terms are based on the relationships between labels, the differences between them, and various distance metrics ($\mathcal{D}_s$ and $\mathcal{D}_a$) applied to state and action pairs. The final expression for the gradient involves further calculations and summations over different combinations of state-action pairs ($k$ and $l$), as well as the products of reward differences, predicted labels, and the distances between state-action pairs. The equation introduces a constant term $C$ which is attributed to the goal of training the graph $G$ to minimize the difference between predicted and true labels.

## A.2  THE PROOF OF REWARD INFERENCE

In this section, we present the proof process for Formula (7). The inference of rewards for unlabelled nodes requires considering the information transfer between labelled and unlabelled nodes, we use

$W_{UL}R_L$ for this calculation, while also taking into account the relationships between unlabeled nodes themselves, computed using $W_{UU}R_U$. Since unknown rewards $R_U$ are a variable to be learned, we provide an iterative computation formula by:

$$R_U \longleftarrow W_{UU}R_U + W_{UL}R_L. \tag{9}$$

*Proof.* Let $W_{UL}R_L = \alpha$, then we have

$$R_U^1 \longleftarrow W_{UU}R_U^0 + \alpha, \tag{10}$$

where $R_U^0$ is the initial $R_U$, and $R_U^1$ is the result of first iteration. Given the second iteration,

$$R_U^2 \longleftarrow W_{UU}(W_{UU}R_U^0 + \alpha) + \alpha, \tag{11}$$

$$R_U^2 \longleftarrow W_{UU}{}^2 R_U^0 + W_{UU}\alpha + \alpha. \tag{12}$$

Given the third iteration,

$$R_U^3 \longleftarrow W_{UU}(W_{UU}{}^2 R_U^0 + W_{UU}\alpha + \alpha) + \alpha, \tag{13}$$

$$R_U^3 \longleftarrow W_{UU}{}^3 R_U^0 + W_{UU}{}^2\alpha + W_{UU}\alpha + \alpha. \tag{14}$$

Given the fourth iteration,

$$R_U^4 \longleftarrow W_{UU}^4 R_U^0 + W_{UU}^3\alpha + W_{UU}^2\alpha + W_{UU}\alpha + \alpha. \tag{15}$$

...... ......

Given the t-th iteration,

$$R_U^t \longleftarrow W_{UU}^t R_U^0 + W_{UU}^{t-1}\alpha + ... + W_{UU}\alpha + \alpha, \tag{16}$$

$$R_U^t \longleftarrow W_{UU}^t R_U^0 + (W_{UU}^{t-1} + ... + W_{UU} + 1)\alpha. \tag{17}$$

$\square$

Based on the weights calculated by Equation (2), the values in $W_{UU}$ are all less than 1. Therefore, as $t$ approaches infinity, $W_{UU}^t$ tends to infinitesimal values, leading $W_{UU}^t R^0$ converges to 0.

Meanwhile, $(W_{UU}^{t-1} + ... + W_{UU} + 1)$ forms a geometric series, and after applying the formula for the sum of a geometric series, we obtain the solution to TRAIN:

$$R_U = (I - W_{UU})^{(-1)} W_{UL} R_L, \tag{18}$$

which is a *fixed point*, and $I$ is the identity matrix (Zhu & Ghahramani, 2002; Zhou et al., 2003).

## A.3 ADDITIONAL EXPERIMENTS

### A.3.1 ABLATION STUDY

We conducted a comprehensive set of ablation studies aimed at thoroughly evaluating the effectiveness of our reward shaping function, denoted as $f_\Theta$. These experiments were meticulously designed and carried out across a diverse set of environments, including four Meta-World environments and five DeepMind Control Suite environments.

In our investigation, we delved into the intricate interplay of various factors that influence the rewards associated with nine distinct tasks. To provide a detailed assessment, we employed four distinct composition methods, each shedding light on the role of factors related to both states and actions:

(1) Method 1: In this approach, both states and actions underwent decomposition into multiple factors, allowing us to scrutinize the combined impact of these nuanced elements.

(2) Method 2: Here, we selectively decomposed states into multiple factors while treating actions as a unified entity. This method offered insights into how states, in isolation, contribute to the shaping of rewards.

(3) Method 3: Conversely, we kept states as a single, undivided factor but decomposed actions into multiple components. This experiment assessed the significance of dissecting actions in the reward-shaping process.

(4) Method 4: As a contrast, we simplified the scenario by considering both states and actions as single, undifferentiated factors. This method served as a baseline for evaluating the performance of more complex factorization approaches.

The experimental results are shown in Table 2.

Table 2: Evaluation returns on four different composition methods for the multiple factors that influence the rewards. The average $\pm$ standard deviation is shown for five random seeds. Italic numbers indicate the highest average return for each task. Bold numbers indicate the statistically significant highest average return for each task, in which the average return is the highest, and there is not a clear overlap in the standard deviation among the baselines.

| Tasks | Method 1 | Method 2 | Method 3 | Method 4 |
|---|---|---|---|---|
| Hammer | **4532 ± 95** | 3158 ± 242 | 2185 ± 262 | 1034 ± 147 |
| Sweep Into | *2257 ± 324* | 1971 ± 243 | 1734 ± 276 | 665 ± 185 |
| Button Press | *3435 ± 106* | 3145 ± 227 | 2576 ± 85 | 1089 ± 290 |
| Open Drawer | **4466 ± 41** | 2998 ± 64 | 2514 ± 54 | 1727 ± 75 |
| Cheetah Run | *430 ± 55* | 361 ± 138 | 255 ± 126 | 183 ± 74 |
| Walker Walk | *950 ± 155* | 714 ± 83 | 463 ± 76 | 262 ± 38 |
| Fish Swim | *576 ± 155* | 453 ± 26 | 296 ± 89 | 198 ± 29 |
| Humanoid Run | **359 ± 57** | 201 ± 31 | 154 ± 38 | 26 ± 4 |
| Cartpole Swingup | **642 ± 68** | 441 ± 29 | 349 ± 18 | 208 ± 17 |

The compelling results that emerged from our extensive experimentation affirmed the superiority of Method 1, where both states and actions were decomposed into multiple factors. This approach consistently demonstrated the most favorable outcomes across the range of tasks we examined.

On the other hand, Method 2, which decomposed states while treating actions as single entities, produced results that, while respectable, fell short of the peak performance achieved by Method 1.

A noteworthy revelation emerged when we explored Method 4, where both states and actions were considered as single factors. This approach exhibited a stark drop in performance, and in some slightly complex environments, it led to outright failures. This finding underscores the critical importance of factorization and highlights the perils of oversimplifying the reward-shaping process.

In light of these insights, we conclude that decomposing both states and actions into multiple factors and seamlessly integrating them using $f_\Theta$ stands as the most effective strategy. This sophisticated approach enables a finer level of granularity in reward learning and significantly enhances the overall efficacy of policies trained on datasets subjected to such comprehensive factorization.

### A.3.2 IMAGE-BASED EXPERIMENTS

Our novel approach, TRAIN, showcases a remarkable degree of adaptability that extends beyond conventional full-state tasks, seamlessly accommodating image-based tasks with equal prowess. In our quest to assess TRAIN's performance in the realm of image-based tasks, we ingeniously conditioned the environment's output to generate images, thus opening up exciting possibilities for visual-based learning scenarios.

Furthermore, we integrated a widely endorsed strategy employed in diverse task domains, wherein we treated multiple frames as a single time-step state. These frames underwent a meticulous process

of decomposition into interdependent frames, and each frame was subsequently subjected to further dissection into RGB channels, effectively treating each single image as an individual factor.

Our experimentation with TRAIN spanned a diverse set of environments, encompassing four distinct Meta-World environments and an additional five environments sourced from the esteemed DeepMind Control Suite. The breadth of this evaluation allowed us to gain comprehensive insights into TRAIN's capabilities in a variety of settings. The results are shown in Figure 7.

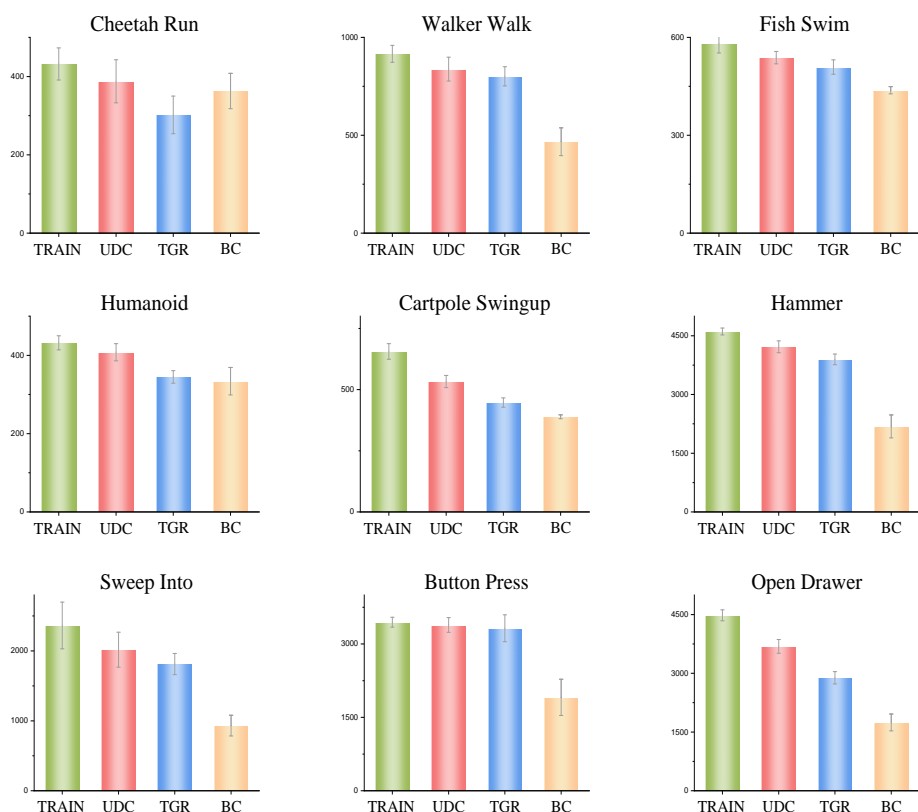

Figure 7: Evaluation returns on the nine image-based tasks. The vertical lines depict the standard deviation across five random seeds of each experiment.

The experimental results that emerged from these rigorous trials underscored TRAIN's robust performance in image-based experiments. This underscores the effectiveness of our approach in seamlessly integrating multiple images within a single time step, each image serving as a distinct factor. This multi-faceted approach capitalizes on the richness of information inherent in each image, ultimately enhancing the depth and quality of the learning process. TRAIN's ability to harness the unique information contained within each image not only expands its versatility but also positions it as a promising candidate for a wide array of image-driven applications in the realm of artificial intelligence.

## A.4 ACCURATE OF PREDICTED LABELS ON DIFFERENT LABELLED DATA PROPORTION

To rigorously assess the predictive accuracy of our innovative approach, TRAIN, across varying ratios of labeled data, we embarked on a comprehensive evaluation campaign. Our experiments spanned a diverse range of environments, encompassing four challenging Meta-World scenarios and an additional five environments sourced from the prestigious DeepMind Control Suite.

In Figure 8, we present a vivid representation of our findings, utilizing the mean squared error (MSE) as our primary evaluation metric. This heatmap graphically depicts the relationship between multiple tasks (on the horizontal axis) and the ratio of reward-labeled data in the dataset (on the vertical axis).

Each value in the figure reflects the MSE associated with a specific task under the corresponding labeled data ratio. To ensure the robustness of our findings, we partitioned the dataset into multiple batches, with the calculated result representing the average MSE value across these batches.

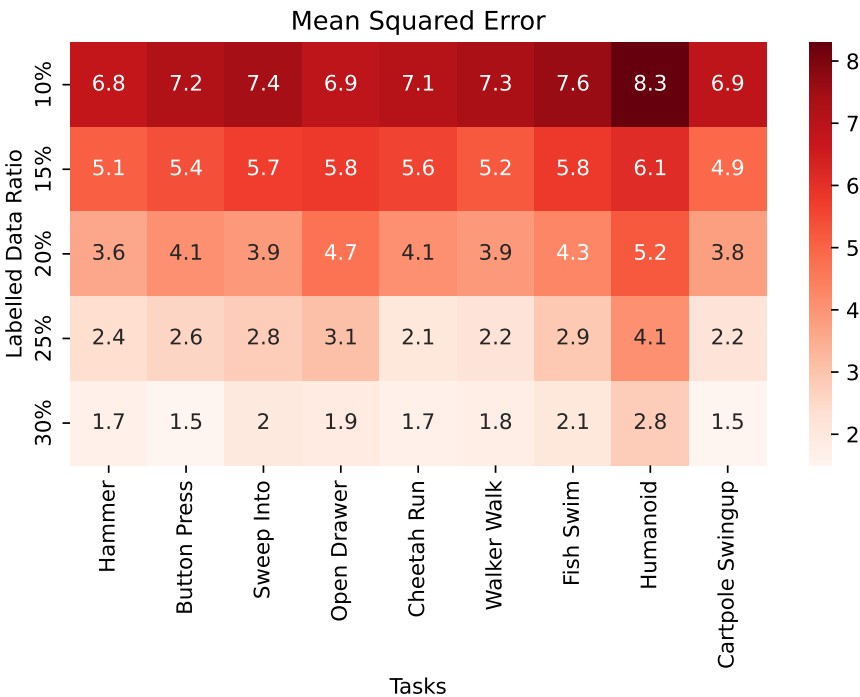

Figure 8: The accuracy between predicted labels and ground truth labels of TRAIN under different labelled data ratios on four Meta-World and five DeepMind Control Suite environments, respectively. The evaluation metric is the mean squared error (MSE).

The compelling insights derived from our experiments reveal a clear trend: as the ratio of reward-labeled data in the dataset increases, the corresponding MSE values decrease, indicating a higher degree of predictive accuracy. Conversely, a lower ratio of labeled reward data in the dataset is associated with higher MSE values, signifying a relatively lower predictive accuracy. Furthermore, our observations indicate that tasks characterized by higher-dimensional states and actions tend to exhibit elevated MSE values, suggesting that predictive challenges are more pronounced in these complex settings.

These findings offer valuable insights into the performance of TRAIN across a spectrum of labeled data ratios and task complexities, shedding light on its capabilities and areas for potential refinement. Such detailed evaluations are instrumental in understanding the nuances of our approach's predictive accuracy and provide a roadmap for its application in real-world scenarios across various domains.

## A.5 ACCURATE OF PREDICTED LABELS ON DIFFERENT NORMS

To assess the accuracy of our proposed method, TRAIN, in relation to different norms, we conducted experiments using four Meta-World and five DeepMind Control Suite environments. Figure 8 illustrates the results, utilizing the mean squared error (MSE) as the evaluation metric.

It's important to note that the 1.5 norm and the 2.5 norm were introduced purely for experimentation purposes and lack specific physical interpretations. These two norms were included to investigate the impact of norm selection on our method.

In the heatmap of the experimental outcomes, the horizontal axis represents various tasks, the vertical axis corresponds to different norms, and the values within the figure indicate the MSE for each task

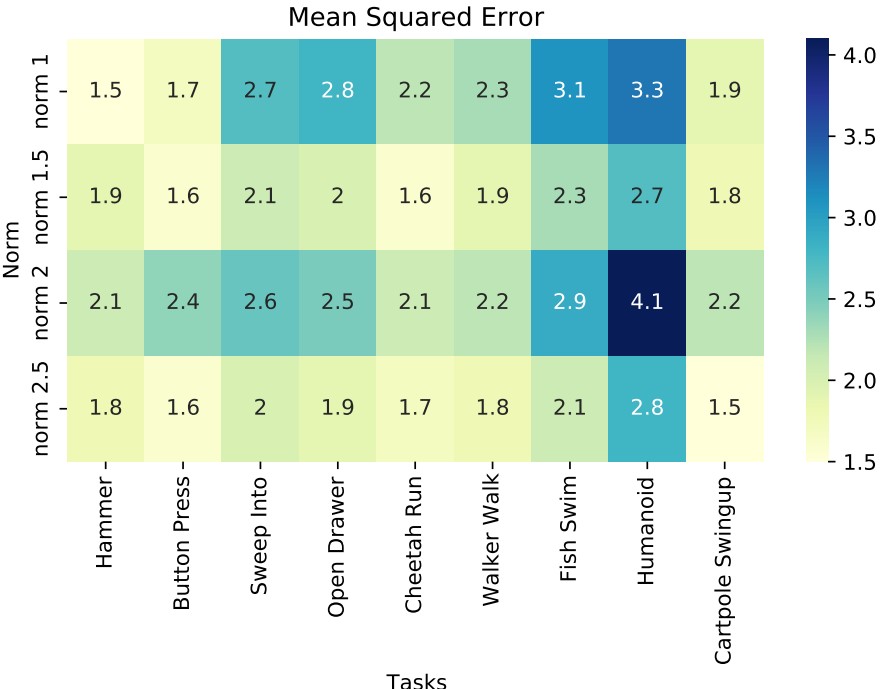

Figure 9: The accuracy between predicted labels and ground truth labels of TRAIN under different norms on four Meta-World and five DeepMind Control Suite environments, respectively. The evaluation metric is the mean squared error (MSE).

under a specific norm. The dataset was partitioned into multiple batches, and the calculated result represents the average MSE across these batches.

From the experimental results, it is evident that different norms have minimal influence on the MSE, indicating that our method TRAIN is not sensitive to the choice of norm, underscoring its robustness in various scenarios.

## A.6 DIFFERENT TOTAL PAIRS AND LABELLED PROPORTION FOR EACH TASK

In Table 3, we show, for each task, the total number (total) of state-action pairs used to train policies and the ratio of state-action pairs labelled rewards. From the data presented in the table, we can observe that for the majority of tasks, using 10% to 15% of data with rewards is sufficient to achieve completion. Only a small number of complex tasks require 20% of data with rewards to accomplish the task.

## A.7 DISCUSSION

### A.7.1 COMPUTATIONAL COST

In this section, we choose to evaluate the computational costs of our method in several experimental environments, including both training time and memory consumption.

Most of our datasets consist of $1 \times 10^6$ state-action pairs, equivalent to $1 \times 10^6$ nodes in the graph. For computational convenience, we segment the dataset. Our dataset comprises training data from classical reinforcement learning algorithms, such as SAC (Soft Actor-Critic). As the policy learns, different "regions" of the dataset exhibit diversity, with "similar" data typically found within the same segment. Therefore, we have reason to believe that slicing the dataset sequentially is a reasonable approach.

Table 3: For each task, the total number (total) of state-action pairs used to train policies and the ratio of state-action pairs labelled rewards.

| Tasks | Total pairs | Labelled data ratio |
|---|---|---|
| basketball | $1 \times 10^6$ | 15% |
| box-close | $1 \times 10^6$ | 10% |
| button-press | $1 \times 10^6$ | 15% |
| button-press-topdown | $1 \times 10^6$ | 15% |
| button-press-topdown-wall | $2 \times 10^6$ | 15% |
| coffee-button | $1 \times 10^6$ | 10% |
| coffee-pull | $1 \times 10^6$ | 15% |
| dial-turn | $1 \times 10^6$ | 15% |
| disassemble | $2 \times 10^6$ | 15% |
| door-close | $1 \times 10^6$ | 10% |
| door-lock | $1 \times 10^6$ | 10% |
| door-open | $1 \times 10^6$ | 15% |
| door-unlock | $1 \times 10^6$ | 15% |
| drawer-close | $1 \times 10^6$ | 10% |
| drawer-open | $1 \times 10^6$ | 10% |
| faucet-close | $1 \times 10^6$ | 10% |
| faucet-open | $1 \times 10^6$ | 10% |
| hammer | $2 \times 10^6$ | 10% |
| hand-insert | $2 \times 10^6$ | 15% |
| handle-press | $1 \times 10^6$ | 15% |
| handle-press-side | $1 \times 10^6$ | 10% |
| handle-pull | $1 \times 10^6$ | 10% |
| handle-pull-side | $1 \times 10^6$ | 10% |
| lever-pull | $1 \times 10^6$ | 10% |
| pick-out-of-hole | $2 \times 10^6$ | 20% |
| pick-place | $2 \times 10^6$ | 15% |
| plate-slide | $1 \times 10^6$ | 10% |
| plate-slide-back | $1 \times 10^6$ | 10% |
| plate-slide-back-side | $1 \times 10^6$ | 10% |
| plate-slide-side | $1 \times 10^6$ | 10% |
| push | $1 \times 10^6$ | 15% |
| push-back | $1 \times 10^6$ | 15% |
| push-wall | $1 \times 10^6$ | 15% |
| reach | $1 \times 10^6$ | 10% |
| reach-wall | $1 \times 10^6$ | 10% |
| stick-pull | $2 \times 10^6$ | 20% |
| stick-push | $2 \times 10^6$ | 20% |
| sweep | $1 \times 10^6$ | 15% |
| sweep-into | $1 \times 10^6$ | 15% |
| window-close | $1 \times 10^6$ | 10% |
| window-open | $1 \times 10^6$ | 10% |
| Cheetah Run | $1 \times 10^6$ | 10% |
| Walker Walk | $1 \times 10^6$ | 10% |
| Fish Swim | $1 \times 10^6$ | 15% |
| Humanoid Run | $1 \times 10^6$ | 20% |
| Cartpole Swingup | $1 \times 10^6$ | 10% |

We conducted tests for dataset slices containing 10000 state-action pairs and for dataset slices containing 20000 state-action pairs on a computer with the following specifications: CPU: Intel i9-9900KF

3.6GHz, GPU: RTX 2070 SUPER (8GB VRAM). The training time and memory consumption for the environments are presented in Table 4 and Table 5.

Table 4: Computational cost for dataset slices containing 10000 state-action pairs.

| Env | Cheetah Run | Walker Walk | Hammer | Door | Pick-place |
|---|---|---|---|---|---|
| Training time | 12.68s | 12.67s | 2.67s | 2.7s | 2.62s |
| Memory consumption | 2351MB | 2351MB | 2377MB | 2377M | 2377MB |

Table 5: Computational cost for dataset slices containing 20000 state-action pairs.

| Env | Cheetah Run | Walker Walk | Hammer | Door | Pick-place |
|---|---|---|---|---|---|
| Training time | 14.8s | 13.7s | 13.07s | 13.9s | 13.52s |
| Memory consumption | 5869MB | 5869MB | 5886MB | 5886M | 5886MB |

We also conducted tests for dataset slices containing 30000 state-action pairs and for dataset slices containing 40000 state-action pairs on a server with the following specifications for larger dataset slices: CPU: Intel Xeon Gold 6230 2.10GHz, GPU: RTX 3090 (24GB VRAM). The training time and memory consumption for the environments are presented in Table 6 and Table 7.

Table 6: Computational cost for dataset slices containing 30000 state-action pairs.

| Env | Cheetah Run | Walker Walk | Hammer | Door | Pick-place |
|---|---|---|---|---|---|
| Training time | 37.61s | 38.07s | 23.12s | 24.34s | 26.72s |
| Memory consumption | 11511MB | 11511MB | 11583MB | 11583M | 11583MB |

Table 7: Computational cost for dataset slices containing 40000 state-action pairs.

| Env | Cheetah Run | Walker Walk | Hammer | Door | Pick-place |
|---|---|---|---|---|---|
| Training time | 39.1s | 40.09s | 28.34s | 29.21s | 29.86s |
| Memory consumption | 19577MB | 19577MB | 19691MB | 19691MB | 19691MB |

We anticipate significantly faster computation on server-grade hardware and the ability to use larger slices on GPUs with greater VRAM.

### A.7.2 TREND IN THE CHANGES OF REWARD INFERENCE VALUES

We examine the changes in the maximum value of $R_U$ to verify whether the learned rewards are bounded by the annotated data. Since $R_U$ is calculated through Formula (7), we design an illustration to demonstrate this issue concerning the formula. We fix $R_L$ as a $3 \times 1$ matrix with values $\{1, 2, 3\}$. Meanwhile, for demonstration purposes, we treat $(I - W_{UU})$ and $W_{UL}$ in Formula (7) as two variables, with values ranging from $0.01$ to $0.09$. The trend of the maximum value in $R_U$ with the variation of $(I - W_{UU})$ and $W_{UL}$ is illustrated in Figure 10.

From the graph, it can be observed that within certain ranges of $(I - W_{UU})$ and $W_{UL}$ values, the maximum value in $R_U$ can exceed the maximum value in $R_L$, that is the learned rewards are not bounded by the annotated data.

### A.7.3 ANOTHER BASELINE

In the preceding sections, we compared our method with reward learning baselines in the field of offline reinforcement learning. The primary concept of $\Phi_{GCN}$ (Klissarov & Precup, 2020) in online reinforcement learning reward learning can also be applied to this scenario.

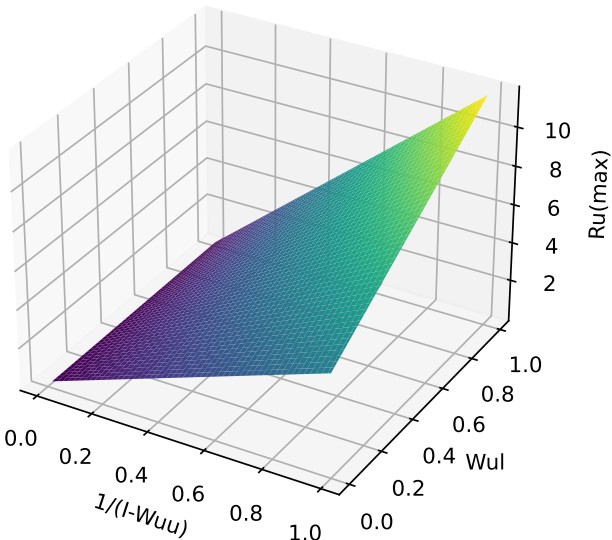

Figure 10: Trend in the changes of reward inference values.

Upon this baseline, we conducted experimental validations on selected environments. In the experiments, we utilized the same data annotation ratio as our approach for $\Phi_{GCN}$. We performed five random seed experiments for each environment, calculating the average and standard deviation of the results. The experimental results are presented in Table 8 and Table 9:

Table 8: Evaluation returns on the 4 DeepMind Control Suite tasks. The average $\pm$ standard deviation is shown for five random seeds.

| Env | Cheetah Run | Walker Walk | Fish Swim | Cartpole Swingup |
|---|---|---|---|---|
| TRAIN | $430 \pm 55.9$ | $952.7 \pm 155.2$ | $567.6 \pm 155.2$ | $642.4 \pm 68.9$ |
| $\Phi_{GCN}$ | $274 \pm 71.1$ | $634 \pm 172.3$ | $181 \pm 93.2$ | $396 \pm 132.1$ |

Table 9: Evaluation returns on the 5 Meta-World tasks. The average $\pm$ standard deviation is shown for five random seeds.

| Env | Coffee Button | Push Wall | Hammer | Open Drawer | Window-open |
|---|---|---|---|---|---|
| TRAIN | $4128 \pm 210.9$ | $4501 \pm 204.6$ | $4532 \pm 59.1$ | $4466 \pm 41.3$ | $3829 \pm 208.4$ |
| $\Phi_{GCN}$ | $2845 \pm 303.8$ | $3238 \pm 192.1$ | $2933 \pm 81.8$ | $2989 \pm 79.3$ | $2921 \pm 234.1$ |

$\Phi_{GCN}$ considers information derived from the state, and therefore performs well in environments where the state contributes significantly to the reward. For the environments where the action has a substantial impact on the reward, the performance of $\Phi_{GCN}$ is slightly inferior.

### A.7.4 BASELINE DETAILS

The Time-guided reward (TGR) workflow, as outlined in (Konyushkova et al., 2020), involves several key steps. Initially, it infers a reward function based on limited supervision, utilizing timestep-level annotations expressed as reward values on a subset of trajectories. Subsequently, it retroactively annotates all trajectories using the obtained reward function. Finally, the trajectories, now equipped with predicted rewards, are employed for offline reinforcement learning. Additionally, we integrated TGR with CRR (Critic Regularized Regression) (Wang et al., 2020) to learn policies as a baseline of our efforts. Specifically, TGR employs a two-step process for reward function inference. It starts by annotating demonstrated trajectories, assigning a flat zero synthetic reward to the unlabelled subset. The reward function is then trained using a loss function that jointly optimizes timestep-level annotations and synthetic flat labels. This comprehensive approach contributes to the effectiveness of TGR in the context of offline reinforcement learning.

The prevailing characteristic of most offline reinforcement learning datasets is the prevalence of a small fraction of labeled data alongside a more substantial proportion of unlabeled data. Unlabeled data sharing (UDS) (Yu et al., 2022) addresses this scenario by treating the unlabeled dataset as if it has zero reward, followed by the incorporation of reweighting techniques. This reweighting process is designed to adjust the distribution of interspersed zero-reward data. The primary objective is to synchronize the distribution of this external data with that of reward-containing data pertinent to the original task, thereby alleviating the bias introduced by inaccurate reward data. In particular, UDS initially assigns the minimum feasible reward (typically assumed to be 0) to all transitions within the unlabeled data. Subsequently, these unlabeled transitions undergo reweighting, altering the distribution of unlabeled data to mitigate reward bias. This strategy contributes to the overall goal of enhancing the alignment between labeled and unlabeled data distributions in UDS. Finally, train offline reinforcement learning policies using the reweighted reward distribution dataset.

