# OpenReview forum: "Offline Reward Inference on Graph: A New Thinking"
_ICLR.cc/2024/Conference — Submitted to ICLR 2024_

### Official Review · Reviewer_EMLT · 2023-10-25

**Soundness:** 3 good
**Presentation:** 3 good
**Contribution:** 3 good
**Rating:** 6
**Confidence:** 2

**Summary:**

The paper proposed a graph based method that infers the rewards of unlabelled state-action pairs for offline RL tasks. The method is tested on several robotic environments and the empirical results show that the proposed algorithm outperforms the existing methods.

**Strengths:**

The paper has clear presentation and the overall idea is easy to understand. The paper performs sufficient empirical experiments and the results are convincing.

**Weaknesses:**

1. The paper doesn't present complete details of the experiment setup. While $f_\Theta$ is an important component for constructing the graph weights, throughout the paper no details on the setup of $f_\Theta$. The paper also doesn't reveal any details about the policy formulation or any parameters related with the training process.

2. While the paper does sufficient comparison over several existing methods, one additional thing that might be worth presenting is comparison with different reward inference methods. For example, one might want compare the paper's method with reward inference simply by KNN.

**Questions:**

1. One potential limit of the method is the size of the weight matrix grows with the number of data points. In this case, it could be very costly to compute the matrix inverse or other related things. Can the author give some comments on the computation cost?

2. An alternative way to infer the reward is to simply use the known part of the reward, i.e., $R_U = W_{UL}R_L$. What's the advantage of the proposed method, compared with this simpler formulation?

---

> ### Author Response · Authors · 2023-11-20
> **To Reviewer EMLT**
>
> I am truly appreciative of your dedication in reviewing my work. Your attention to detail and the suggestions you provided have immensely contributed to its improvement.
>
> ### [I]. Explanation of weaknesses
>
> > **[1/2] W1:** The paper doesn't present complete details of the experiment setup.
>
> For most environments, the network architecture of the function $f_\Theta$ is a two-layer fully connected network with ReLU activation, and the number of hidden layer neurons is 64.
>
> The network structures for policy training are as follows:
>
> For Meta-World environments, it is a two-layer fully connected network with ReLU activation and the number of hidden layer neurons is 64.
>
> For DeepMind Control Suite environments, it is a three-layer neural network with ReLU activation and the number of hidden layer neurons is 64.
>
> > **[2/2] W2:** While the paper does sufficient comparison over several existing methods.
>
> As shown in Table 3 of Appendix A.5, our data has annotations for only a small fraction (mostly 10%-15%) of the environments. Therefore, for deep learning fully supervised methods, we found them to be ineffective in our testing. We believe that the KNN-based approach is a promising idea and may yield good results for certain problems. However, applying the KNN method in this scenario requires complementary measurement methods and an appropriate way of propagating rewards. We will give special consideration to the suggestions you have provided in our future work.
>
> ### [II]. Explanation of questions
>
> > **[1/2] Q1:** Can the author give some comments on the computation cost?
>
> Thanks for the insightful question. For this question, we choose to evaluate the computational costs of our method in several experimental environments, including training time and memory consumption. We also include the results as a section (A.7.1) in the paper's Appendix.
>
> Most of our datasets consist of $1 \times 10^6$ state-action pairs. For computational convenience, we segment the dataset. We conducted tests on a computer with the following specifications: CPU: Intel i9-9900KF 3.6GHz, GPU: RTX 2070 SUPER (8GB VRAM). The training time and memory consumption for several environments are presented below:
>
> Dataset slices containing 10000 state-action pairs:
> |         Env         | Cheetah Run | Walker Walk | Hammer | Door  | Pick-place |
> |:-------------------:|:-----------:|:-----------:|:------:|:-----:|:----------:|
> |    Training time    |   12.68s    |   12.67s    | 2.67s  | 2.7s  |   2.62s    |
> | Memory  consumption |   2351MB    |   2351MB    | 2377MB | 2377M |   2377MB   |
>
> Dataset slices containing 20000 state-action pairs:
> |         Env         | Cheetah Run | Walker Walk | Hammer | Door  | Pick-place |
> |:-------------------:|:-----------:|:-----------:|:------:|:-----:|:----------:|
> |    Training time    |    14.8s    |    13.7s    | 13.07s | 13.9s |   13.52s   |
> | Memory  consumption |   5869MB    |   5869MB    | 5886MB | 5886M |   5886MB   |
>
> We also conducted tests on a server with the following specifications for larger dataset slices: CPU: Intel Xeon Gold 6230 2.10GHz, GPU: RTX 3090 (24GB VRAM).
>
> Dataset slices containing 30000 state-action pairs:
> |         Env         | Cheetah Run | Walker Walk | Hammer  |  Door  | Pick-place |
> |:-------------------:|:-----------:|:-----------:|:-------:|:------:|:----------:|
> |    Training time    |   37.61s    |   38.07s    | 23.12s  | 24.34s |   26.72s   |
> | Memory  consumption |   11511MB   |   11511MB   | 11583MB | 11583M |  11583MB   |
>
> Dataset slices containing 40000 state-action pairs:
> |         Env         | Cheetah Run | Walker Walk | Hammer |  Door  | Pick-place |
> |:-------------------:|:-----------:|:-----------:|:------:|:------:|:----------:|
> |    Training time    |    39.1s    |   40.09s    | 28.34s | 29.21s |   29.86s   |
> | Memory  consumption |   19577MB   |   19577MB   | 19691MB | 19691MB  |   19691MB   |
>
>
> > **[2/2] Q2:** What's the advantage of the proposed method, compared with this simpler formulation?
>
> Thanks for this insightful question. We have carefully considered it and modified the corresponding section in the paper (Section 4.3). Additionally, we have included the detailed derivation process in the Appendix (A.2).
>
> The inference of rewards for unlabelled nodes requires considering the information transfer between labelled and unlabelled nodes, we use $W_{UL}R_L$ for this calculation while also taking into account the relationships between unlabeled nodes themselves, computed using $W_{UU}R_U$.
> Since unknown rewards $R_U$ are a variable to be learned, we provide an iterative computation formula by:
> \begin{equation}
> R_U \longleftarrow W_{UU}R_U + W_{UL}R_L.
> \end{equation}
>
> -------------
> We once again sincerely thank Reviewer EMLT from the bottom of our for reviewing our paper. We hope our answers have addressed all your concerns. If so, we would greatly appreciate it if Reviewer EMLT could consider raising their score. Please let us know if there are more questions.

---

### Official Review · Reviewer_jJQM · 2023-10-31

**Soundness:** 3 good
**Presentation:** 2 fair
**Contribution:** 3 good
**Rating:** 6
**Confidence:** 4

**Summary:**

The paper considers the offline RL setting where reward labels are scarce. In possible application of offline RL, such labels are difficulty/unethical to obtain (such as in healthcare), therefore we need to find ways to maximize the knowledge contained in the available reward labels. The authors propose to construct a graph out of the available data and perform reward inference on nodes for which no reward is available. The method is evaluated on continuous control problems, such as DeepMind Control and Meta World.

**Strengths:**

* The method proposes an intuitive solution to the problem of reward scarcity in the offline RL setting
* The presentation of the paper is good and understandable
* The method is evaluated across a wide range of environments

Learning efficiently in offline RL necessarily depends on having reward information for each state action pair. When that is not the case practitioners have to find ways to infer the missing rewards labels. The paper proposes a way to infer such labels by considering the underlying geometry of the problem together with considerations with respect to different factors that influence the reward.

**Weaknesses:**

* The baselines used are not well presented. Closely related work is not compared to
* The empirical evaluation relies on only 5 seeds. Moreover, the presentation of the results does not consider statistical significance
* No investigation is done to understand the impact of different choices in the algorithm

The TGR and UDS are presented too succinctly. If the reader is not familiar with these methods it is impossible to understand how they relate. For example "UDS relabels unlabeled data with zero rewards"  seems like a very strange way to deal with unlabeled data - it essentially doesn't deal with it. More details would be needed.

The paper proposes to "learn a reward propagation graph and infer rewards", yet a quick google search on "reward propagation" reveals a paper from 2020 that proposes a similar method to infer rewards on nodes with missing reward labels [1]. Although their method is for the online RL setting, it seems straightforward to use it in the offline RL setting. This would be an important baseline.

The empirical evaluation is not adequate from a scientific rigour point-of-view. Only 5 seeds are used and Table 1 highlights the score of methods for which the standard deviation intersects with other methods. This is misleading, only non-intersecting standard deviations should be highlighted. Perhaps additional seeds could help in this.

Very little work is done to understand the impact of different components in the method. For example, what is the impact of using a reward propagation graph? Why not simply infer the reward through a classification/regression loss that takes as input only the current state. This is another important baseline. Finally what is the impact the different reward factors?

**Questions:**

Section 4.4 "we" typo -> it should be capitalized

"we regard each part of the state and action as a factor that influences the reward" Doesn't this add more prior knowledge into the problem? What happens if we don't know the factors?

Figure 2 "prpagation" typo

"Cabi et al. (2020) is hard to sketch tasks" perhaps this needs some rephrasing

"Konyushkova et al. (2020) assumes rewards are binary, not adaptation to any value reward learning question." This could perhaps also use some rephrasing.


================================================

[1] Reward Propagation Using Graph Convolutional Networks. Klissarov and Precup. 2020

---

> ### Author Response · Authors · 2023-11-20
> **To Reviewer jJQM**
>
> Thanks, Reviewer jJQM, for generously sharing your time and knowledge in reviewing our work. Your professionalism and attention to detail have played a crucial role in enhancing the quality of our paper.
>
> ### [I]. Explanation of weaknesses
>
> > **[1/4] W1:** The TGR and UDS are presented too succinctly.
>
> Thanks for pointing out this crucial issue. We have provided further details in the description of TGR and UDS.
>
> > **[2/4] W2:** Reward Propagation Using Graph Convolutional Networks would be an important baseline.
>
> $\Phi_{GCN}$ is an outstanding and impactful work. We have cited the mentioned article in our paper, and the citation has been updated in the PDF. Upon this baseline, we conducted experimental validations on selected environments. In the experiments, we utilized the same data annotation ratio as our approach for $\Phi_{GCN}$. We performed five random seed experiments for each environment, calculating the average and standard deviation of the results. The experimental results are presented in the following table:
>
> |  Env  |  Cheetah Run   |    Walker Walk    |     Fish Swim     | Cartpole Swingup |
> |:-----:|:--------------:|:-----------------:|:-----------------:|:----------------:|
> | TRAIN | 430 $\pm$ 55.9 | 952.7 $\pm$ 155.2 | 567.6 $\pm$ 155.2 | 642.4 $\pm$ 68.9 |
> |  $\Phi_{GCN}$  | 274 $\pm$ 71.1 |  634 $\pm$ 172.3  |  181 $\pm$ 93.2   | 396 $\pm$ 132.1  |
>
>
> |  Env  |  Coffee Button   |    Push Wall     |     Hammer      |   Open Drawer   |   Window-open    |
> |:-----:|:----------------:|:----------------:|:---------------:|:---------------:|:----------------:|
> | TRAIN | 4128 $\pm$ 210.9 | 4501 $\pm$ 204.6 | 4532 $\pm$ 59.1 | 4466 $\pm$ 41.3 | 3829 $\pm$ 208.4 |
> |  $\Phi_{GCN}$  | 2845 $\pm$ 303.8 | 3238 $\pm$ 192.1 | 2933 $\pm$ 81.8 | 2989 $\pm$ 79.3 | 2921 $\pm$ 234.1 |
>
> $\Phi_{GCN}$ considers information derived from the state and performs well in environments where the state contributes significantly to the reward. For the environments where the action substantially impacts the reward, the performance of $\Phi_{GCN}$ is slightly inferior.
>
> > **[3/4] W3:** The empirical evaluation is not adequate from a scientific rigour point-of-view.
>
> 1) The goal of RL can be described by maximizing the expected cumulative reward.
> In this setting, environments exhibit randomness, and different initializations of these environments result in varying learned policies, which affects the std value of Meta-World tasks.
> For statistical convenience, the average of returns in those tasks presents an intuitive observation of the policies' performance. It is convinced that the std values that assist in showing the stability of the policies in random environments are additional statistics.
>
> 2) In the reinforcement learning community, adopting 5 seeds is a feasible setting. In our experimental setup, conducting experiments with 10 seeds would take approximately 4 to 6 months, which is a substantial time investment.
>
> 3) Based on your suggestion, we have highlighted the statistically significant highest average returns for each task with bold numbers, in which the average return is the highest, and there is not a clear overlap for the standard deviations among the baselines. For the other results, we only mark the highest average returns for each task with italic numbers, which avoids confusion caused by intersecting standard deviations. We have modified Table 1 and Table 2 in the draft, including updating the captions for both tables.
>
> > **[4/4] W4:** Very little work is done to understand the impact of different components in the method.
>
> The information in our ablation study in Appendix A.2.1 could address your concern. We evaluated the performance of methods employing four different input modes. We endeavored to utilize deep learning (supervised learning) for reward learning, but deep learning requires substantial amounts of annotated data, and our dataset comprises at most 10%-20% annotated data. This quantity is inadequate for deep learning, making the approach based on deep learning (supervised learning) practically ineffective.
>
> ### [II]. Explanation of questions
>
> > **[1/4] Q1:** Section 4.4 "we" typo -> it should be capitalized
>
> Thank you very much, we have made the necessary corrections.
>
> > **[2/4] Q2:** Doesn't this add more prior knowledge into the problem?
>
> Thanks for the insightful question. We believe that if the factors' information is known, it achieves a fine-tuning of the input. If uncertain, one can directly learn along the dimensions of state and action.
>
> > **[3/4] Q3:** Figure 2 "prpagation" typo
>
> Thank you very much, we have updated the content of the images accordingly.
>
> > **[4/4] Q4-5:**  some rephrasing
>
> The two modifications you suggested are extremely valuable. Your suggestions prompted us to reconsider related content carefully. We have modified the paragraph and updated it accordingly.
>
> ---
> We once again sincerely thank Reviewer jJQM for reviewing our paper.

---

> > ### Comment · Reviewer_jJQM · 2023-11-22
> >
> > Dear authors,
> >
> > thank you for your detailed feedback and the additional baseline. The paper overall looks more positive to me.
> >
> > I still feel like the TGR and UDS baselines are not explained deeply enough. Could the authors add more details in the appendix?
> >
> > My comment about scientific rigour is not really about theoretical guarantees but more from the point of view of having a proper evaluation. This means having more than 5 seeds (10 minimum) and being rigorous in how results are presented. Right now Table 1 still highlights in bold results that are not statistically significant. What I mean is that the average score is higher, but there is clear overlap in the standard deviation. This should be fixed.
> >
> > Thank you for Appendix A.3.1, this has helped understand the contribution. Please also be make sure not to highlight statistically insignificant results in Table 2.

---

> > > ### Author Response · Authors · 2023-11-23
> > > **Response to Reviewer jJQM**
> > >
> > > Dear Reviewer jJQM,
> > >
> > > Thank you for your valuable feedback.
> > >
> > > I sincerely apologize for the misunderstanding of your question.
> > >
> > > 1) Reinforcement Learning is based on the idea of the reward hypothesis, all goals can be described by the maximization of the expected cumulative reward.
> > > In this setting, environments exhibit randomness, and different initializations of these environments result in varying learned policies, which affects the std value of Meta-World tasks.
> > > For statistical convenience, the average of returns in those tasks presents an intuitive observation of the policies' performance. It is convinced that the std values that assist in showing the stability of the policies in random environments are additional statistics.
> > >
> > > 2) In the reinforcement learning community, adopting 5 seeds is a feasible setting, such as the recently publications[1][2][3][4][5][6].
> > >
> > > 3) In our experimental setup, our computational conditions include a server with the following specifications for larger dataset slices: CPU: Intel Xeon Gold 6230 2.10GHz, GPU: RTX 3090 (24GB VRAM). Conducting experiments with 10 seeds would take approximately 4 to 6 months, which is a substantial time investment.
> > >
> > > 4) Based on your suggestion, we have highlighted the statistically significant highest average returns for each task with bold numbers, in which the average return is the highest, and there is not a clear overlap for the standard deviations among the baselines. For the other results, we only mark the highest average returns for each task with italic numbers, which avoids confusion caused by intersecting standard deviations. We have modified Table 1 and Table 2 in the draft, including updating the captions for both tables.
> > >
> > > [1] Confidence-Conditioned Value Functions for Offline Reinforcement Learning. Joey Hong, et al. ICLR 2023.
> > >
> > > [2] In-sample Actor Critic for Offline Reinforcement Learning. Hongchang Zhang, et al. ICLR 2023.
> > >
> > > [3] Supported Trust Region Optimization for Offline Reinforcement Learning. Yixiu Mao, et al.
> > > ICML 2023.
> > >
> > > [4] Policy Regularization with Dataset Constraint for Offline Reinforcement Learning. Yuhang Ran, et al. ICML 2023.
> > >
> > > [5] One Risk to Rule Them All: A Risk-Sensitive Perspective on Model-Based Offline Reinforcement Learning. Marc Rigter, et al. NeurIPS 2023.
> > >
> > > [6] Efficient Diffusion Policies for Offline Reinforcement Learning. Bingyi Kang, et al. NeurIPS 2023.
> > >
> > > ===========================================================================
> > >
> > > Based on your suggestion, we added more details about TGR and UDS baselines in the Appendix (A.7.4).

---

### Official Review · Reviewer_kgGb · 2023-10-31

**Soundness:** 3 good
**Presentation:** 4 excellent
**Contribution:** 3 good
**Rating:** 6
**Confidence:** 4

**Summary:**

This study introduces a reward inference technique designed for offline Reinforcement Learning (RL) scenarios in which only a subset of the available data are annotated with reward information. This approach leverages a graph-based framework to extend reward inference to unannotated data points. In this approach, a neural network is trained to predict the weights between nodes in the graph, facilitating the propagation of reward information. Empirical evaluations demonstrate that this approach yields superior reward quality compared to existing baseline methods.

**Strengths:**

* This paper offers a theoretical foundation by providing guarantees that the proposed reward inference method can converge to at least a local optimum, which enhances its credibility and applicability.

* The manuscript is well-written and presents the research in a clear and easily understandable manner.

* The problem addressed in this work is importance, as it tackles a critical challenge in offline reinforcement learning when applying to real world situation, enabling its practical application in situations where traditional methods might fall short.

**Weaknesses:**

See Questions

**Questions:**

* In the paper, the reward shaping function $f_{\theta}$ is trained using annotated data to convert state-action pairs into scalar values, and this information is used to calculate the weights between nodes in the graph. Since $f_{\theta}$ must learn the relative importance of each dimension in the node data, it raises the question of the necessary diversity within the annotated data. For instance, if all the annotated data consists of expert transitions with the highest rewards, it might be challenging for the function to discern which dimensions of the node data contribute most significantly to changes in reward. Hence, an important question to address is: "What level of diversity within the annotated data is required for the proposed method to learn an effective function $f_{\theta$?"
* Is the reward calculated by the graph method bounded by the range of rewards in the annotated data, i.e., $\max{(R_U)}\leq \max{(R_L)}$? This question seeks to understand the extent to which the graph-based reward inference can maintain the upper bound of reward values, as derived from the annotated data.
* To make the proposed method practically feasible, it's essential to investigate the computational cost requirements. Specifically, showing the computational time and memory consumption required for reward inference helps provide insights into the practical applicability and efficiency of the method in real-world scenarios.

---

> ### Author Response · Authors · 2023-11-20
> **To Reviewer kgGb**
>
> I am genuinely appreciative of your dedication to reviewing my work. Your valuable questions and insights have significantly enriched the quality of our work. We briefly summarize the questions into the following three questions:
>
> > **[1/3] Q1:** How diverse is enough for the proposed method to learn a decent function $f_\Theta$?
>
> Thanks for the very valuable and profound question. The commonly used practice in existing work involves datasets generated by training classic reinforcement learning algorithms (such as SAC) policies online[1][2]. During the online training process, the policy evolves from weak to strong, interacting with the environment and generating diverse data. The question raised by the reviewer is very valuable; it represents a specific case but is indeed a scenario that could exist. We ensure the diversity of our dataset in two ways. Firstly, our dataset also adopts the training process data of classic reinforcement learning algorithms (such as SAC). Secondly, we use uniform sampling on the dataset to label state-action pairs, ensuring the uniform presence of annotated state-action pairs with rewards in various "regions" of the dataset. In our future work, we will consider how to address the specific case raised by the reviewer.
>
> [1] How to Leverage Unlabeled Data in Offline Reinforcement Learning. Yu, Tianhe, et al. ICML 2022.
>
> [2] Conservative Data Sharing for Multi-task Offline Reinforcement Learning. Yu, Tianhe, et al. NeurIPS 2021.
>
> > **[2/3] Q2:** Are the rewards calculated by the graph bounded by the annotated data, which means $max(R_U) \leq max(R_L)$?
>
> Thanks for the deepful and valuable question. For this question, we added a section in the paper's Appendix A.7.2 to examine the changes in the maximum value of $R_U$ and verify whether the learned reward values are bounded by the annotated data. We design an illustration to demonstrate this issue concerning the formula, which is also shown in Appendix A.7.2. We can observe that within certain ranges of $(I-W_{UU})$ and $W_{UL}$ values, the maximum value in $R_U$ can exceed the maximum value in $R_L$, that is the learned rewards are not bounded by the annotated data.
>
> > **[3/3] Q3:** What is the computation cost requirement for the method in terms of computation time for the reward inference to converge and memory consumption?
>
> Thanks for the insightful question. For this question, we choose to evaluate the computational costs of our method in several experimental environments, including both training time and memory consumption. We also include the results as a section (A.7.1) in the Appendix of the paper.
>
> The majority of our datasets comprise $1 \times 10^6$ state-action pairs. To facilitate computation, we segment the dataset. We conducted tests on a computer with the following specifications: CPU: Intel i9-9900KF 3.6GHz, GPU: RTX 2070s (8GB VRAM). The training time and memory consumption for several environments are presented below:
>
> Dataset slices containing 10000 state-action pairs:
> |         Env         | Cheetah Run | Walker Walk | Hammer | Door  | Pick-place |
> |:-------------------:|:-----------:|:-----------:|:------:|:-----:|:----------:|
> |    Training time    |   12.68s    |   12.67s    | 2.67s  | 2.7s  |   2.62s    |
> | Memory  consumption |   2351MB    |   2351MB    | 2377MB | 2377M |   2377MB   |
>
> Dataset slices containing 20000 state-action pairs:
> |         Env         | Cheetah Run | Walker Walk | Hammer | Door  | Pick-place |
> |:-------------------:|:-----------:|:-----------:|:------:|:-----:|:----------:|
> |    Training time    |    14.8s    |    13.7s    | 13.07s | 13.9s |   13.52s   |
> | Memory  consumption |   5869MB    |   5869MB    | 5886MB | 5886M |   5886MB   |
>
> We also conducted tests on a server with the following specifications for larger dataset slices: CPU: Intel Xeon Gold 6230 2.10GHz, GPU: RTX 3090 (24GB VRAM).
>
> Dataset slices containing 30000 state-action pairs:
> |         Env         | Cheetah Run | Walker Walk | Hammer  |  Door  | Pick-place |
> |:-------------------:|:-----------:|:-----------:|:-------:|:------:|:----------:|
> |    Training time    |   37.61s    |   38.07s    | 23.12s  | 24.34s |   26.72s   |
> | Memory  consumption |   11511MB   |   11511MB   | 11583MB | 11583M |  11583MB   |
>
> Dataset slices containing 40000 state-action pairs:
> |         Env         | Cheetah Run | Walker Walk | Hammer |  Door  | Pick-place |
> |:-------------------:|:-----------:|:-----------:|:------:|:------:|:----------:|
> |    Training time    |    39.1s    |   40.09s    | 28.34s | 29.21s |   29.86s   |
> | Memory  consumption |   19577MB   |   19577MB   | 19691MB | 19691MB  |   19691MB   |
>
> ---
> We once again sincerely thank Reviewer kgGb from the bottom of our for reviewing our paper and giving suggestions. We hope our answers have addressed all the your concerns. If so, we would greatly appreciate it if Reviewer kgGb could consider raising their score. Please let us know if there are more questions.

---

### Meta-Review · Area_Chair_6adr · 2023-12-10

**Metareview:**

This paper focuses on offline RL and addresses the challenge of reward sparsity. The authors propose a graph-based framework for reward inference, which utilizes available reward labels to infer missing information in unannotated data points. This approach uses a graph-based method to extend reward inference to unannotated data points. A neural net is trained to predict weights between nodes in the graph to propagate rewards between them. The key evaluation metric is expected rewards across a range of environments.

The paper includes empirical evaluations in various environments, including DeepMind Control and Meta World. Additionally, it provides a some theoretical groundings, discussing the method's potential for convergence to at least a local optimum.

All reviewers agreed that this is an interesting and important direction to pursue. However, the key question that remained unanswered was regarding robust empirical evaluations. In particular, concerns about statistical significance (number of seeds) and baselines (e.g. \phi_GCN). The authors ran some additional baseline results but due to shortage of time/compute, they said they could not run additional experiments in time for the reviewers to be convinced. So I believe a more robust eval is missing and I would encourage the authors to do another round of iteration before resubmitting the paper.

**Justification For Why Not Higher Score:**

Need more experimental validation to better ground the contributions

**Justification For Why Not Lower Score:**

N/A

---

### Decision · Program_Chairs · 2024-01-16

Reject